# Design and Fabrication of a Triple-Band Terahertz Metamaterial Absorber

**DOI:** 10.3390/nano11051110

**Published:** 2021-04-25

**Authors:** Jinfeng Wang, Tingting Lang, Zhi Hong, Meiyu Xiao, Jing Yu

**Affiliations:** 1Institute of Optoelectronic Technology, China Jiliang University, Hangzhou 310018, China; S1904080311@cjlu.edu.cn (J.W.); P20040854085@cjlu.edu.cn (M.X.); yujing362898577@163.com (J.Y.); 2Centre for THz Research, China Jiliang University, Hangzhou 310018, China; hongzhi@cjlu.edu.cn

**Keywords:** a triple-band, metamaterial absorber, electric and magnetic fields, polarization independent

## Abstract

We presented and manufactured a triple-band terahertz (THz) metamaterial absorber with three concentric square ring metallic resonators, a polyethylene terephthalate (PET) layer, and a metallic substrate. The simulation results demonstrate that the absorptivity of 99.5%, 86.4%, and 98.4% can be achieved at resonant frequency of 0.337, 0.496, and 0.718 THz, respectively. The experimental results show three distinct absorption peaks at 0.366, 0.512, and 0.751 THz, which is mostly agreement with the simulation. We analyzed the absorption mechanism from the distribution of electric and magnetic fields. The sensitivity of the three peaks of this triple-band absorber to the surrounding is 72, 103.5, 139.5 GHz/RIU, respectively. In addition, the absorber is polarization insensitive because of the symmetric configuration. The absorber can simultaneously exhibit high absorption effect at incident angles up to 60° for transverse electric (TE) polarization and 70° for transverse magnetic (TM) polarization. This presented terahertz metamaterial absorber with a triple-band absorption and easy fabrication can find important applications in biological sensing, THz imaging, filter and optical communication.

## 1. Introduction

Terahertz wave is usually defined as radiation at frequencies between 0.1 and 10 THz [1]. Terahertz technology is of significant interest to researchers and has a wide range of potential applications, including radar cross-section reduction [2], wireless communications [3], selective thermal emission [4], and THz spectroscopy [5]. Metamaterials, with periodic subwavelength structures, have attracted considerable attention owing to their novel physical properties [6,7], such as near zero index [8], cloaking [9], cross polarization conversion [10], and perfect absorption [11]. In order to regulate the electromagnetic wave, a large number of metamaterial devices have been confirmed. In particular, metamaterial absorbers have become a research hotspot because of their perfect absorption performance [12,13,14]. Metamaterial absorbers are designed for various devices, such as spatial light modulators [15], thermal emitters [16], multiplexing detector arrays [17], sensors [18,19], etc. In 2008, the first metamaterial absorber with a metal-dielectric composite structure was proposed, which exhibited near-unity absorption at microwave frequency [20]. Subsequently, various metamaterial absorbers with excellent performance have been reported in the visible [21], infrared [22], terahertz [23], and microwave bands [24]. Among them, the absorber working in various bands can achieve single band [23], multiband [24], and broadband absorption [1].

At present, one research hot spot of the absorber structure tends to realize multiband absorption [25,26,27]. A common strategy is to include multiple resonators with different-size in a unit cell [28,29]. This method can achieve multiband absorption because of multiple resonance combinations of continuous resonance frequencies. For example, Zhang et al. [17] presented a multispectral absorber using four metal squares of different sizes in a unit cell, which exhibited a triple-band absorption. Zheng et al. [30] designed a four-band metamaterial absorber based on flower-shaped patterns of different sizes in a unit cell. However, the arrangement of multiple resonant elements greatly increases the size of the structure, which does not meet the requirements of the current integrated development. At present, the unit structure composed of metal resonant elements with different shapes has become a new design direction [25]. For example, an absorber that exhibited five-band peak absorption was reported; it consisted of different metallic resonators on top of a dielectric spacing layer and a gold ground plane [25]. Qin et al. [31] investigated a triple-band absorber based on three different gold particles binding. Metal resonators with different shapes are used to obtain multiband absorption, which greatly simplifies the structure. Not only metal-based absorbers, but also other absorber structures can achieve good absorption performance. For example, a graphene-based THz absorber [32] was proposed, which realized a double broadband absorption. Michael. A.C. at al. [33] propose an all dielectric metamaterial absorber; the structure achieved a narrow near-perfect absorption in the THz band. The absorber based on graphene is complex in structure, and the absorber based on all dielectric has higher requirements for the fabrication accuracy. With the development of lithography technology, the practical application of absorbers based on metal resonators becomes possible. As one of the core technologies of micro nano processing, lithography is facing the challenges of high cost, complex process and low resolution. To tackle the challenges for scalability of lithographic approaches, a template-assisted colloidal self-assembly approach has been proposed as an alternative to fabricate macroscopic magnetic metasurfaces [34]. The approach induces the gold nanorods to excite a magnetic resonance, which has application values towards the colloidal fabrication of functional optical metasurfaces [34]. Nowadays, various multiband absorbers have been theoretically proposed, but few have been confirmed experimentally, limiting the further application value.

In this paper, we designed and manufactured a triple-band terahertz metamaterial absorber using a combination of three concentric square ring metallic resonators. The absorber shows three narrowband absorption peaks at frequencies of 0.337, 0.496, and 0.718 THz with absorption rates of 99.5%, 86.4%, and 98.4%, respectively. The experimental results are mostly in agreement with simulations. We also explored the absorption mechanism from the distribution of electric and magnetic fields. In addition, the absorption characteristics are polarization insensitive owing to the structural symmetry. The designed structure can simultaneously maintain a high peak absorptivity at incident angles up to 60° for TE polarization and 70° for TM polarization.

## 2. Structure and Design

Figure 1a shows the structure of the proposed metamaterial absorber, which has three parts: three concentric square ring metallic resonators, a polyethylene terephthalate (PET) layer, and a metallic substrate. Figure 1b,c show top and side views of the proposed absorber, respectively. Aluminum, which has a conductivity of 3.56 × 10^7^ S/m, was selected as the metallic resonator and substrate, and the permittivity of PET is 3.2. The unit size of the absorber is set as: *a* = 200 nm, *h* = 10 μm, *t* = 200 nm, *p* = 150 μm, *l*_1_ = 60 μm, *l*_2_ = 90 μm, *l*_3_ = 120 μm, and *w* = 10 μm.

The performance of the proposed absorber was examined using Computer Simulation Technology (CST) Microwave Studio (Computer Simulation Technology Ltd., Darmstadt, Germany). Numerical simulation of absorption for the absorber was calculated using the frequency domain solver provided by the CST software. In the simulation, unit cell boundary conditions were applied in the *x* and *y* directions, and an open boundary condition was applied in the *z* direction to eliminate scattering effects. The incident wave was a plane wave propagating along the negative *z* axis. We define TE (TM) polarization as occurring when the electric (magnetic) field is oriented along the *x* axis. This paper presents the simulation results for TM polarization, as shown on the top of Figure 1a. The absorption (*A*) can be calculated as *A*(*w*) = 1 − *R*(*w*) – *T*(*w*) = 1 − |*S*_11_|^2^ − |*S*_21_|^2^, where *T* and *R* are the transmittance and reflectivity of the absorber, and *S*_11_ and *S*_21_ represent the reflection and transmission coefficients, respectively. Because the metallic substrate of the absorber is thicker enough to block terahertz waves, the transmission is close to zero. Consequently, the absorption can be written as *A*(*w*) = 1 − *R* (*w*) = 1 − |*S*_11_|^2^.

Samples were fabricated according to the following steps, as shown in Figure 2a. A 3 mm-thick quartz wafer was selected as the sample substrate and was cleaned and dried. We used a purchased composite film consisting of a PET layer (13 μm thick) and an aluminum layer (200 nm thick). It should be explained that the simulation in this paper was carried out for PET thickness of 10 μm in the optimal case. Since the PET film that is readily available in the market is 13 μm, we choose this thickness for experiment. Firstly, we applied ethanol to the quartz wafer; we then gently placed the PET-aluminum film onto the quartz wafer with tweezers and pressed away the air between the composite material and the quartz wafer with a cotton swab. After the liquid evaporated, the composite film firmly adhered to the surface of the quartz wafer. Secondly, positive liquid photoresist was spin-coated on the aluminum surface, and baked for 90 s in an oven at 100 °C. Thirdly, photolithography was used to define the geometry of the metamaterials, followed by wet etching (aluminum etching solution, HPO_4_/C_2_H_5_OH/HNO_3_/H_2_O = 16:1:1:3) to fabricate the metamaterial structures. Fourthly, photoresist was washed with acetone and then gently removed from the quartz layer for subsequent processing. Fifthly, a metallic aluminum substrate layer (200 nm) was deposited on the other side of the PET layer by vacuum evaporation. Note that we used a high-temperature adhesive tape to attach the PET material with the metamaterial structure on a hollowed-out mask during metal substrate layer deposition to protect the side with the metamaterial structure. Finally, a 60 × 60 periodic array was fabricated, and the size of this absorber is 9 × 9 mm^2^. The reflection spectra were characterized using terahertz time-domain spectroscopy. In the actual measurements, the spot diameter of the light source is 6 mm, and the central area of the array is illuminated. The number of resonators covered by the light source is large enough to maintain stable absorption performance. The experimental results are consistent with the simulation results. An optical microscopy image of a portion of the fabricated absorber structure is shown in Figure 2b.

## 3. Simulation Results and Discussion

The simulated reflectivity, transmittance, and absorptivity curves under normal incidence are shown in Figure 3. The absorber clearly produces three absorption peaks at 0.337, 0.496, and 0.718 THz, with absorption rates of 99.5%, 86.4%, and 98.4%, respectively.

We used spectral decomposition to analyze the three absorption peaks of the structure. Figure 4 shows the absorption spectrum of different square ring resonators, which responds in three frequency bands. Figure 4a–c present the absorption spectra of absorbers consisting of the outer, middle, and inner ring, respectively. In these three cases, the absorber exhibits 99.8%, 97.8%, and 88.9% narrowband absorption at 0.335, 0.48, and 0.709 THz, respectively. We also simulated the absorption spectrum of the entire model, as shown in Figure 4d. The results show three absorption peaks at 0.337, 0.496, and 0.718 THz, where each peak corresponds roughly to the peak of a single ring. There is some spectral mismatch between superposition and individual contributions. The first absorption peak of the entire model is basically consistent with the resonance frequency of the outer resonant ring absorber, but the second and third peaks are not completely match with the absorber consisting of the middle and inner resonant rings. The reason is that the electric and magnetic resonance intensity of the second and third peak changes due to the coupling of these electromagnetic waves. In addition, the interaction between the neighboring triple-square-rings also has some influence on the absorption peak. We can determine that the first, second, and third absorption peaks are generated by the outer, middle, and inner ring, respectively.

In order to evaluate the performance of our absorbers, we compared the results with other structures [31,35,36,37,38], as depicted in Table 1. From the table, we can see that of the three peaks, the absorption rate of the first and third peak is more than 98%. There are also absorbers with higher peak absorption, but their structures are relatively complex, polarization sensitive, and have not been experimentally confirmed. Compared with other absorbers, our structure has the advantage of being polarization insensitive, easy to fabricate, and having a thin dimension and a simple construction.

**Table 1 nanomaterials-11-01110-t001:** Comparison results of various three-band absorbers.

Ref.	Structure	Unit Size (μm)	Waveband (THz)	Peak Numbers	The First Peak Absorption	The Second Peak Absorption	The Third Peak Absorption	Polarization	Experiment
[31]	Au resonators-SiO_2_-Au	0.6	100–300	3	96.8%	99.6%	99.2%	insensitive	No
[35]	Au resonators-Si-Au	200	0.1–1	3	97.6%	96.5%	84.1%	insensitive	No
[36]	Al resonators- polyimide -Al	300	0.15–0.85	3	80%	81%	79%	sensitive	Yes
[37]	Au resonators-GaAs-Au	30	1–6	3	99.4%	99.6%	98.2%	insensitive	No
[38]	Au resonators- dielectric- Au resonators- dielectric- Au resonators- dielectric-Au	60	0.8–3.7	3	Nearly 100%	Nearly 100%	Nearly 100%	sensitive	No
This work	Al resonators -PET-Al	150	0.1–1	3	99.5%,	86.4%	98.4%	insensitive	Yes

The distributions of the electric and magnetic fields are presented to illustrate the mechanism of absorption in the three bands in more detail. Figure 5a–c show the distributions of the electric field in the *x*–*y* plane, and Figure 5d–f show the distributions of the magnetic field in the *x*–*y* plane at the three resonance frequencies of 0.337, 0.496, and 0.718 THz, respectively. As shown in Figure 5a,d, the electric and magnetic fields are concentrated mainly on the outer ring and its edge, indicating strong electrical and magnetic resonances [23,39]. The distribution of the electric and magnetic fields proves that the first absorption peak is excited by the outer ring resonator. Figure 5b,e show that at the second peak frequency, 0.496 THz, electric and magnetic fields are focused primarily on the middle ring and its edge, indicating strong electrical and magnetic resonances. At the resonance frequency of 0.718 THz, the electric and magnetic fields are distributed mainly on the inner ring and its edge, as shown in Figure 5c,f, respectively, demonstrating that the third absorption peak is excited by this resonator. In order to explain the absorption mechanism more clearly, the distributions of electric and magnetic fields in the case of three combinations of configurations with two rings (middle + inner, outer + inner, outer + middle) at the resonance frequency 0.337, 0.496, and 0.718 THz are investigated in Figure 5g–l, respectively. As can be seen from Figure 5g,j, when there are only middle and inner resonators, the electric and magnetic fields at the first resonance frequency decrease significantly. However, both the middle and inner rings have the phenomenon of electric and magnetic fields enhancement, but the effect is not obvious and does not cause strong electric and magnetic resonances. The phenomena indicates that the first peak is mainly the outermost excitation. The mechanism of the second and third resonant peaks is the same the first peak as shown in Figure 5h,k,i,l. The above analysis verifies that the electric and magnetic resonances of the three concentric square rings produce three-band absorption.

We also explored the effects of various parameters on the absorption characteristics. First, we adjusted the thickness of the PET layer while keeping other parameters unchanged. Figure 6a shows the simulated absorption spectra for four different thicknesses values of PET layer. As lossless dielectric, the PET layer is related to the coupling between the bottom aluminum layer and the aluminum rings. As the thickness *h* increases from 10 to 19 μm, the intensity of the first and third absorption peaks decreases, whereas that of the second absorption peak first increases and then remains constant. The change of electric and magnetic resonance intensity leads to the change of the second absorption peak amplitude. When the impedance of the absorber matches the impedance of the free space, the absorption will reach to 100% [40]. The increased thickness *h* of PET enhanced the impedance mismatch of the first and the third peak between the absorber and free space, resulting in different peak shift [40]. Next, we examined the effect of the width (2, 4, 6, 8, and 10 μm) of the metallic resonator on the absorption. Figure 6b clearly shows that when the width of the concentric square ring resonator is decreased from 10 to 2 µm, the three absorption peaks red-shifted, and the absorption tends to decrease. The decrease in the area of the ring resonators weakens the electric and magnetic resonance, which results in a decrease in the peak absorption.

To evaluate the sensing performance of this triple-band narrowband absorber, the dependence of the absorption spectrum on the refractive index (*n*) change of the surrounding is presented in Figure 7. The refraction index varies from 1 to 1.6 and increases at intervals of 0.2. As shown in Figure 7a, when the *n* of the surrounding is changed from 1 to 1.6, the three peak frequencies all redshifted. For sensing, sensitivity (*S*) is a significant factor to describe the sensing performance, and the sensitivity *S* is defined as: *S* = Δ*f/*Δ*n*, where Δ*f* and Δ*n* are the changes of the peak frequency and the refractive index, respectively [36]. Figure 7b–d show the fitting results of the three peak’s frequency with the corresponding *n*. The sensitivity of the three peaks to the external refractive index are 72, 103.5, 139.5 GHz/RIU, respectively. As shown in Table 2, we compare the sensitivity to the refractive index of this proposed absorber with some reported results [12,36,41]. Our values are slightly higher than some of them. The above results prove the feasibility of the proposed absorber for sensing applications [18,36].

**Table 2 nanomaterials-11-01110-t002:** Sensitivity comparison results of various THz absorbers.

Ref.	Waveband (THz)	S(GHz/RIU)/Peak 1	S(GHz/RIU)/Peak 2	S(GHz/RIU)/Peak 3
[12]	0.5–3	1150	3050	-
[36]	0.15–0.85	54.18	119.2	139.2
[41]	0.5–4.5	83	100	125
This work	0.1–1	72	103.5	139.5

The polarization performance of the designed structure under TE and TM polarization is shown in Figure 8a,b, respectively. The simulated results clearly present that the intensity of the three absorption peaks is independent of the polarization angle *φ* under normal incidence. The reason is the perfect symmetry of the structure. In fact, the light beam is oblique incident to the device in most cases, so it is of great significance to design an absorber with the performance of angle insensitive. As shown in Figure 8c,d, absorption maps with incident angles *θ* varying from 0 to 70° for TE and TM polarization are presented. When the incident angle *θ* increases from 0 to 60°, the first absorption peak maintains 80% absorption, and the third absorption peak exhibits >80% absorption in Figure 8c. For TM polarization, when the incident angle is increased to 70°, the absorption efficiency of the three peaks is maintained. For TE polarization, the peak absorption is gradually reduced with the increase of incident angle. The reason is that the direction of the electric field varies with the incident angle, which leads to the decrease of the electric resonance intensity [23,39]. However, there is no change in direction of the electric field in TM polarization, under which the resonator maintains high absorptivity [23,39] for incident angles up to 70°, as shown in Figure 8d. The results in Figure 8c,d show that when the incident angle exceeds 10°, two additional weak absorption peaks are observed in TE and TM polarization. The extra resonance peak is caused by the higher resonance mode, which has been confirmed in experiments [4,42,43]. The higher resonance mode is generated by resonance within the dielectric and occurs at shorter wavelengths with a slightly narrower absorption peak at oblique incident [4,43]. 

## 4. Experiment and Results

In this study, we manufactured four types of absorber structures: resonators consisting of an outer, middle, or inner square ring or three concentric square rings. In our measurement, firstly, we reduce the humidity of the test environment to 6%; then we paste the film of the absorber on the bracket of the measurement system; finally, the reflection spectrum is calculated. Figure 9 shows the simulated and measured absorption spectra of each absorber. Microscopic images of the first three absorber structures are shown next to the spectra in Figure 9a–c. The fabricated outer ring absorber exhibits an absorption peak at approximately 0.385 THz, as shown in Figure 9a, with 66.4% absorption. The measured spectrum for the middle ring resonator in Figure 9b shows 74.7% absorption at 0.531 THz, which is slightly different from the simulation results. As shown in Figure 9c, the inner ring absorber exhibits 88.6% absorption at 0.796 THz, which exceeds the simulated peak absorption of 69.1%. Although there are minor differences in amplitude and absorption bandwidth (which can probably be attributed to imperfections, aluminum oxidation, and measurement errors), the measured spectra is basically consistent with the trends predicted by numerical simulations. The slight frequency difference between the experiment and the simulation should be attributed to the size error and the round corners of the square ring resonators obtained in the manufacturing process. The simulated peak frequencies of the entire absorber structure are 0.337, 0.496, and 0.718 THz, as shown in Figure 9d. The measurements show three absorption peaks at 0.366, 0.512, and 0.751 THz, which is in reasonable agreement with the simulation results. In addition, as the PET substrate is very soft owing to its thickness (approximately 13 μm), there is some undesired deformation in the fabricated samples, and the surface flatness of the structure also has some effect on the absorption. Because of the existing experimental conditions, the spectral quality is not ideal. We are trying to improve the signal-to-noise ratio of the measurement system in the aspects of increasing the signal collection time and restraining noise to improve the spectral quality.

## 5. Conclusions

In conclusion, we designed and manufactured a triple-band terahertz metamaterial absorber consisting of three concentric square ring metallic resonators on top of a PET layer and a metallic substrate. Simulation results showed that the resonators exhibit three absorption peaks of 99.5%, 86.4%, and 98.4% at 0.337, 0.496, and 0.718 THz, respectively. The experimental results verified the reliability of the simulation results. The electric and magnetic field distributions at the three resonance frequencies were explored to examine the peak absorption. This proposed absorber provides the feasibility for sensing applications. The absorber can maintain 80% peak absorptivity at incident angles up to 60° for TE polarization and >80% peak absorptivity for incident angles up to 70° for TM polarization. The advantages of our absorber include a thin size, polarization independence, multiband absorption, and incident angle insensitivity. The use of PET raw material can provide a reference for the future production of flexible absorbers. This study supports applications in terahertz sensing, imaging, plasma-enhanced photoelectric devices, and other areas.

## Figures and Tables

**Figure 1 nanomaterials-11-01110-f001:**
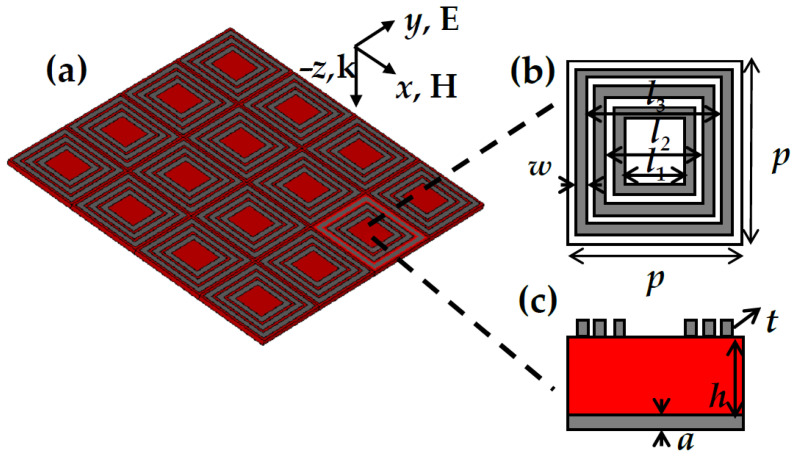
(**a**) Three-dimensional structure diagram of the absorber; (**b**) top and (**c**) side views of unit cell structure. The unit size is set as: *w* = 10 μm, *p* = 150 µm, *l*_1_ = 60 µm, *l*_2_ = 90 µm, *l*_3_ = 120 µm, *t* = 200 nm, *h* = 10 µm, *a* = 200 nm.

**Figure 2 nanomaterials-11-01110-f002:**
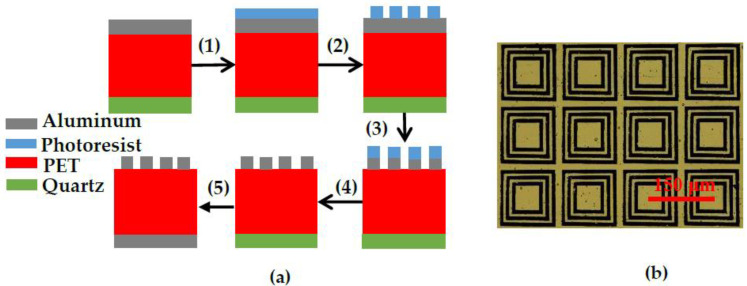
(**a**) Process flow: (1), (2) Photolithography to define the metamaterial geometry, (3), (4) wet etching and washing of the photoresist to fabricate the metamaterial structures, (5) evaporation of metallic substrate layer on other side of PET layer; (**b**) optical microscopy image of fabricated absorber structure.

**Figure 3 nanomaterials-11-01110-f003:**
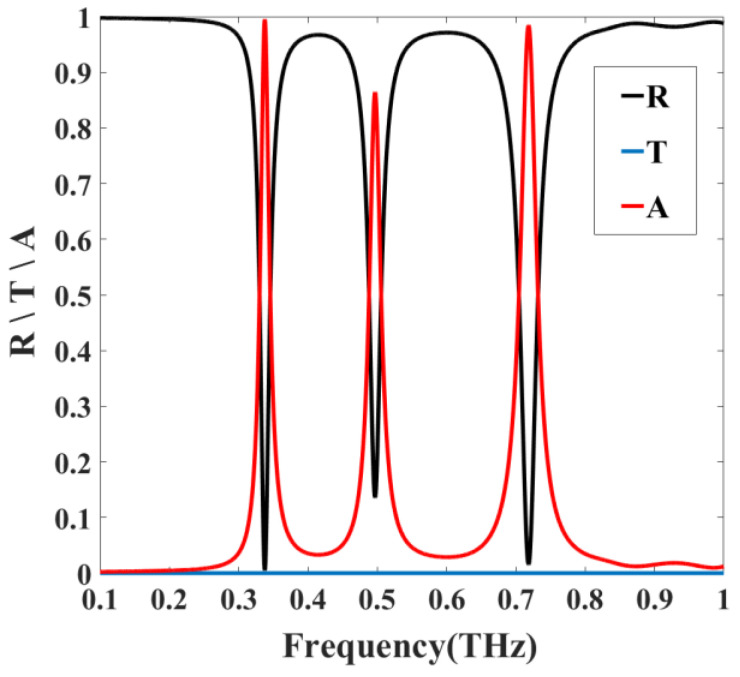
Simulated absorption, reflection, and transmission spectra of the absorber.

**Figure 4 nanomaterials-11-01110-f004:**
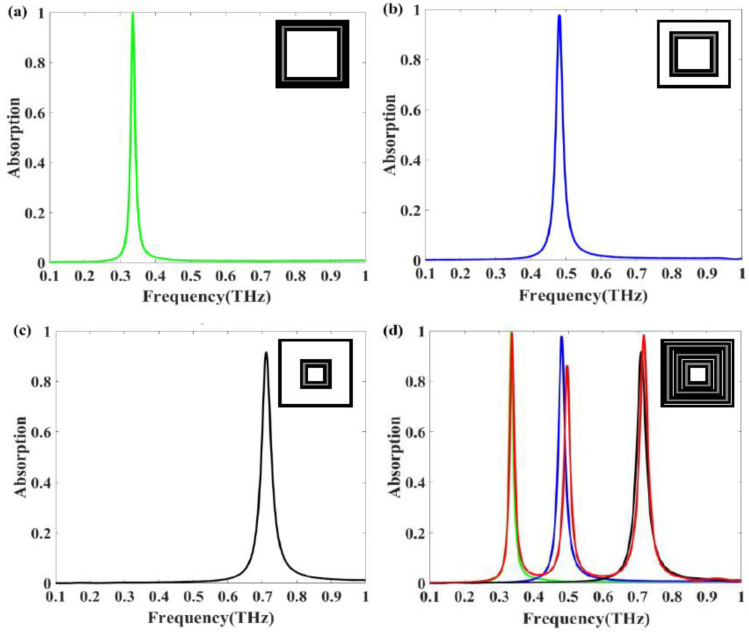
Simulated absorption spectra of absorbers: (**a**) outer square ring resonator; (**b**) middle square ring resonator; (**c**) inner square ring resonator; (**d**) resonator with three concentric square rings.

**Figure 5 nanomaterials-11-01110-f005:**
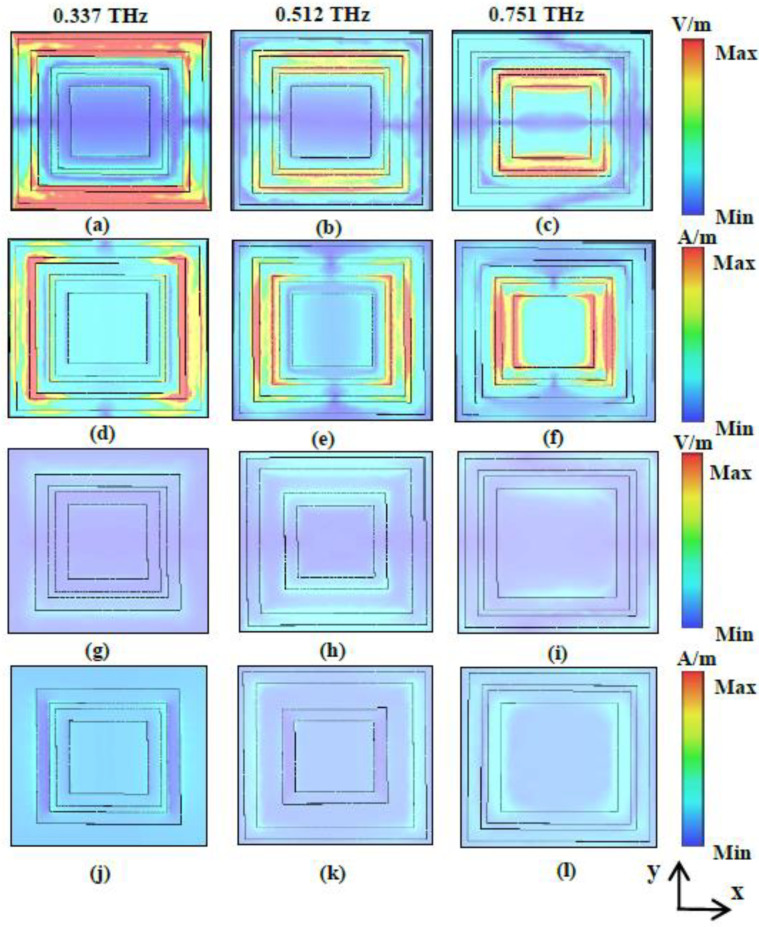
Electric field distributions in *x*–*y* plane of the unit cell at resonance frequencies of (**a**) 0.337 THz; (**b**) 0.496 THz; (**c**) 0.718 THz and magnetic field distributions in *x*–*y* plane of the unit cell at resonance frequencies of (**d**) 0.337 THz; (**e**) 0.496 THz; (**f**) 0.718 THz; electric field distributions in *x*–*y* plane of the unit cell in the case of three combinations of configurations with two rings (middle + inner, outer + inner, outer + middle) at the resonance frequencies (**g**) 0.337; (**h**) 0.496; (**i**) 0.718 THz and magnetic field distributions in *x*–*y* plane of the unit cell at the resonance frequencies (**j**) 0.337; (**k**) 0.496; (**l**) 0.718 THz.

**Figure 6 nanomaterials-11-01110-f006:**
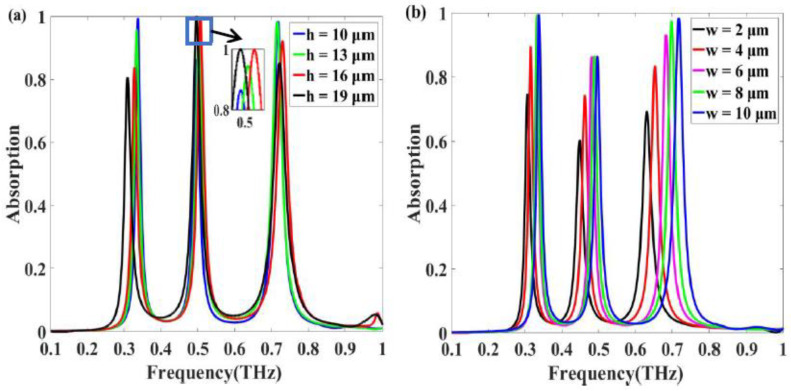
Simulated absorption spectra of the absorber with (**a**) four different thickness values *h* of PET layer; (**b**) five different widths *w* of resonators consisting of three concentric square rings.

**Figure 7 nanomaterials-11-01110-f007:**
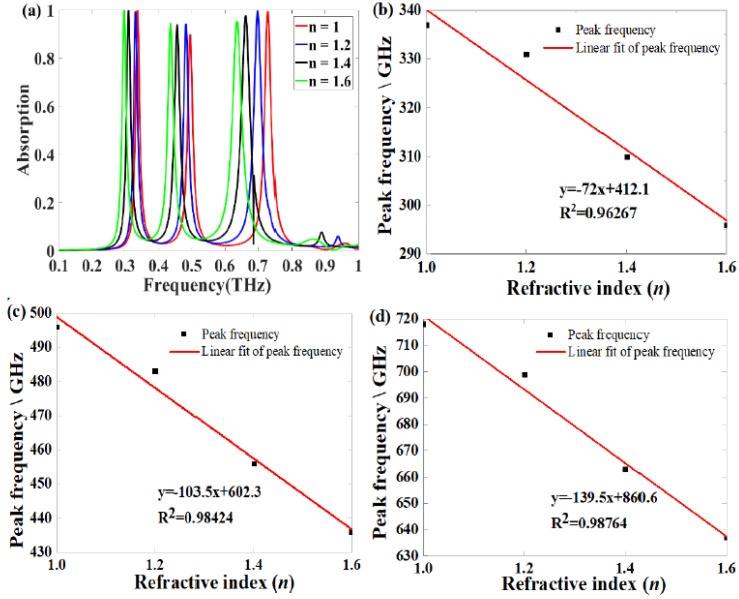
(**a**) Dependence of the absorption spectrum on the changes of the refractive index (*n*) of the surrounding; (**b**–**d**) corresponding linear fit of the three peaks’ frequency versus the corresponding *n*.

**Figure 8 nanomaterials-11-01110-f008:**
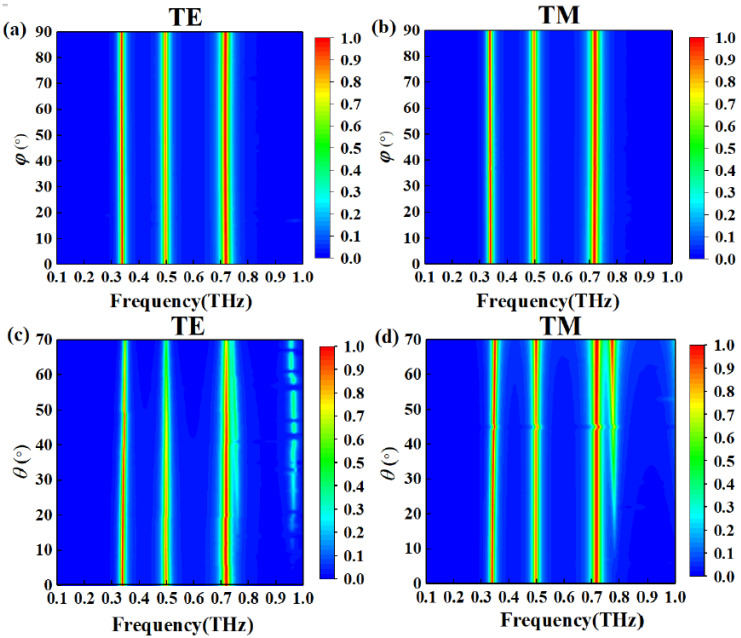
Absorption maps of the absorber under different polarization angles *φ* for (**a**) TE and (**b**) TM polarization; and under different incident angles *θ* for (**c**) TE and (**d**) TM polarization.

**Figure 9 nanomaterials-11-01110-f009:**
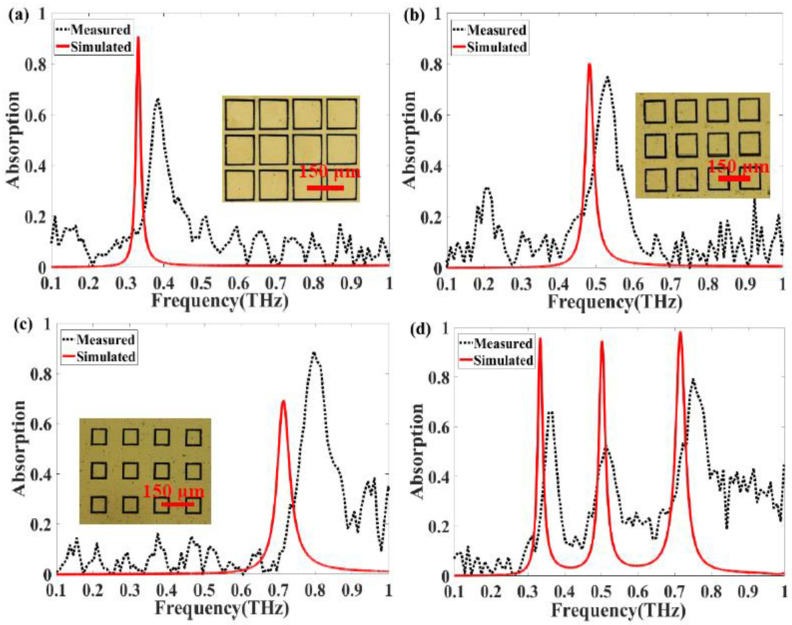
Simulated and measured spectra of absorber structures: (**a**) outer square ring resonator; (**b**) middle square ring resonator; (**c**) inner square ring resonator; (**d**) resonator with three concentric square rings.

## Data Availability

Not applicable.

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
