# Peer review of "Design and Fabrication of a Triple-Band Terahertz Metamaterial Absorber"

_nanomaterials, 2021, doi:10.3390/nano11051110_

Round 1
Reviewer 1 Report
This work focuses on a a triple-band terahertz (THz) metamaterial absorber with three concentric square ring metallic resonators.
The experimental results show three distinct absorption peaks at 0.366, 0.512 and 0.751 THz, which in agreement with the simulation. The proposed absorber can simultaneously exhibit high absorption effect at incident angles up to 60° for transverse electric (TE) polarization and 70° for transverse magnetic (TM) polarization.
The presented terahertz metamaterial absorber can be easily fabricated and can find applications in biological sensing, THz imaging, filter and optical communications.
The authors are presenting both simulations and experiments, in a good a good agreement, nevertheless some minor revision is needed.
- Some grammar and syntax issues need to be resolved
- The authors presented a THz metamaterial absorber that can be easily fabricated and can find applications in biological sensing. I strongly suggest the authors to present at least some simulations regarding its sensing possibilities. Use 2 or 3 realistic cases and present the data.
This work can be published in MDPI Nanomaterials after cover the above minor issues.
Reviewer 2 Report
This paper describes a triple-band terahertz metamaterial absorber consisting of three concentric square rings metallic resonators on top of a PET layer and a metallic substrate. This paper shows the reliability of the experimental and simulation results. The results may contribute to future research on flexible absorbers. It is also thought that it can be applied to various fields.
With the revision, the manuscript can be considered for publication in nanomaterials.
Comments:
1) In Fig. 1(b), the unit size suggested by the author is different from the size indicated in the figure. I think l2 should be greater than 50um. Check it again, please.
2) On Page 2 line 79, the authors say “This paper presents the simulation results for TM polarization, as shown Fig.1” but Fig.1 is a structure diagram of the absorber.
3) In Table 1, the author compared the results with other structures in Table1. The advantage of the structure designed in this study is not clear. (ex. How different are the dimensions?) Please indicate in more detail.
4) On Page 5 line 161, I think the thicknesses values of PET is h, not t. Check it again.
5) In Fig. 5(a), The second peak shifted differently from the first and third peaks. What is the reason?
6) On Page 7 line 215, Why did the author choose 13um for PET thickness?
Reviewer 3 Report
Jinfeng Wang and coauthors report on meta-absorbers based on concentric closed square-ring resonators by a combination of experimental and theoretical methods. The focus of this work is the design of triple-band absorption on the THz regime. I very much enjoyed the combination of the rational design by theoretical consideration followed by the experimental realization. Overall, I consider this contribution ideally suited for the journal nanomaterials. However, few issues have become apparent during reading and I would welcome the authors to address these in a revision before acceptance for publication can be recommended. Nevertheless, this is a very well presented study and I am optimistic that these issues listed below can be resolved within additional simulations and text revisions.
Comments/questions:
- Many studies have observed that the number of resonators included in the excitation volume is of high importance. Thus, which size was the illuminated area and what is the number of resonators covered by the excitation volume”?
- To tackle the challenges for scalability of lithographic approaches, colloidal approaches have been proposed as an alternative to fabricate macroscopic magnetic metasurfaces (Mayer et al. Faraday Discussions 2016, 191, 159–176). Therefore, I want to invite the authors to expand their discussion on the state-of-the-art concepts to fabricate metasurfaces with magnetic resonances.
- Structural symmetry was indicated to be an important factor. Did the authors consider to fabricate anisotropic “rectangular”-ring resonators? Is there an application scenario where such anisotropic resonators (multi-band/broad-band) might become of interest?
- Figure 2 gives the simulated absorption, reflection, and transmission spectra. The authors might wish to clarify for the broad audience to which extend “scattering” contributions come into play. Are such diffuse radiative contributions included in the reflection and if, how about diffuse scattering? This needs a brief clarification in the main text.
- Figure 3 shows the deconvolution/superposition of individual resonator contributions. What is the main factors that could explain the spectral mismatch between superposition and individual contributions?
- In Figure 4 (a and d), the inner ring is only weakly involved on the field enhancements. Is it truly decoupled? It might be helpful to also simulate the three possible combinations of configurations with two rings (large+medium, medium+small, large+small) and to compare the result with the three-ring results. This would improve the comprehensibility of the design aspect of the theoretical discussion.
- The influence of the PET layer on the mode position is not clear. Is this related to a coupling between the bottom gold layer and the gold rings? Please clarify in the manuscript.
- Minor typo in Figure 7 a) “Aluminum”.
- Figure 7b: The resonators seem to be situated relatively close to each other. At such narrow positioning, is there an interaction to be expected between the neighboring triple-square-rings? Please clarify in the manuscript.
- In Figure 8, what is the reason for the spectral offset between the simulated and the measured bands?
- The experimental data shown in Figure 8 suffers from poor signal-to-noise ratios. The authors might wish to explain in more detail the measurement conditions and whether increasing the signal collection time would help to improve the “spectral quality”.
Reviewer 4 Report
Here, the authors developed a THz-resonant metasurface to absorb the incident electromagnetic wave across the sub-THz frequencies. Using numerical studies and experimental investigations, it is shown that the proposed metastructure supports substantial absorption at three different frequencies. Although the authors described the spectral properties and validated the claims through a set of analyses, the work suffers from important shortcomings. I listed my concerns below and suggest the authors to address the comments and apply version carefully in the revised version of the manuscript.
General comments:
1) There are obvious grammatical mistakes and typos in the manuscript. A comprehensive polish must be conducted to improve the quality of the writing. For example; "...which are mostly agreement with the simulation" and many other similar mistakes.
2) The colors in the diagrammatically represented schematics must be consistent with the actual hues of the metals. For instance, the employed yellow color has been conventionally used to show gold. Here, grey theme can be used instead.
3) Figure 7 and the corresponding explanations of the lithography procedures can be moved to the beginning of the study.
4) It would be better to distinguish the experimental and numerical studies in the graphs by the use of dashed curves.
Specific comments:
5) In Figure 6d; What does " higher resonance mode" refer to? This needs for technical explanation and detailed discussion.
6) There is 0.1 THz (100 GHz) difference between the experimental and numerical results! It looks like the measurements were accomplished in noisy area, or may be there is an undesired deformation in the fabricated samples. This needs for explanation. Otherwise, the claim of a great agreement between the measured and numerical spectra cannot be used.
7) The simulation details of CST program must be explained further in more details.
8) There must be comparison between different terahertz absorber structures. For instance, it must be indicated that how the developed metallic structure is superior than the graphene-based (Journal of Nanoparticle Research 19(1), 3 (2017)) , and all-dielectric THz metastructures (Nanotechnology, 27(42), 424003 (2016)).
Round 2
Reviewer 3 Report
The authors have provided a revised version and I am almost completely satisfied with the changes. The inclusion of new simulations and the accompanying extended discussion are very well done. I recommend acceptance of the manuscript for publication, but would suggest the authors to proof read the main text and double-check the references, as few typos came to my attention: Line 132: mm should be mm2 (squaremillimeters); Line 425 "Kaya&"; Line 428 missing space in "Huygens’metasurfaces“; Line 430: „Kuttnera,“ should be „Kuttner,“.
Reviewer 4 Report
Acceptable as is.
